# Image-Based Measurement of Wave Interactions with Rubble Mound Breakwaters

**Steven Douglas [1,\*]**, **Andrew Cornett [1,2]** and **Ioan Nistor [1]**

1   Department of Civil Engineering, University of Ottawa, 161 Louis Pasteur, Ottawa, ON K1N 6N5, Canada; andrew.cornett@nrc-cnrc.gc.ca (A.C.); inistor@uottawa.ca (I.N.)
2   Ocean, Coastal, and River Engineering Research Centre, National Research Council of Canada (NRC-OCRE), 1200 Montreal Rd., Ottawa, ON K1A 0R6, Canada
\*   Correspondence: sdoug041@uottawa.ca

**Abstract:** Over the past decade, the use of imaging devices to perform quantitative measurements has seen wide-scale adoption and has become integral to the emerging fields of research, such as computer vision and artificial intelligence. Recent studies, published across a wide variety of fields, have demonstrated a vast number of ways through which image-based measurement systems can be used in their respective fields. A growing number of studies have demonstrated applications in coastal and ocean research. Edge detection methods have been used to measure water surface and bedform elevation from recorded video taken during wave flume experiments. The turbulent mixing of air and water, induced by the breaking waves and the runup processes, poses a particular problem for the edge-detection methods, since they rely on a sharp contrast between air and water. In this paper, an alternative method for tracking water surface, based on color segmentation, is presented. A set of experiments were conducted whereby the proposed method was used to detect water surface profiles for various types of breaking waves interacting with a rubble mound breakwater. The results were further processed to compute the surface velocity during runup. The time-history of surface velocity is shown to closely parallel the point measurements taken nearby the instrumented armor unit. These velocities can potentially serve as boundary conditions for determining the dynamic loads exerted on the armour units. Further, the image processing results are used to remove the time-varying buoyant force from the measured force acting on an individual armour unit, providing additional insight into how the forces develop over time.

**Keywords:** image processing; image-based measurement; wave-structure interaction; rubble mound; breakwater

## 1. Introduction

Measurements of temporal and spatial variation in water surface elevation are fundamental for understanding most ocean and coastal processes. A wide range of techniques for gathering water surface elevation data in, both, field and laboratory applications have been developed over the years. The use of capacitance-type surface-piercing wave probes has been, and remains, standard practice in most hydraulic laboratories. Surface-piercing probes offer the ability to measure changes in water surface elevation over time, along a single line. Multiple probes can be installed in an array for measuring the spatial distribution of water surface elevation. Spatial resolution is generally limited by the minimum distance required to avoid electrical interference between individual probes, typically on the order of tens of millimeters. However, the costs associated with deploying a large number of probes over an area might also be prohibitive. For many laboratory applications, surface-piercing probes are an attractive option for both their accuracy (on the order of 1mm) and relative ease of use.

Surface-piercing probes cannot, however, resolve the ambiguity of multiple water surface locations along the line on which it samples; such is the case for an overturning wave. Significant amounts of entrapped or entrained air in the fluid might also cause the probe to yield inaccurate readings.

The growth of computing power in recent decades has enabled the development and wide-scale application of digital image analysis techniques. Digital image analysis has been adopted in many fields, with applications ranging from medical imaging, robotics, and security. Image analysis techniques are non-intrusive and can provide continuous sampling over large areas at a relatively high spatial resolution. A number of studies have shown image analysis to be a promising tool within the field of ocean and coastal engineering, with many interesting applications. Of particular relevance to the work presented here are the methods that have emerged for measuring water surface elevation from digital images.

In the field of full-scale measurement of ocean waves, recent improvements to stereo image processing algorithms has enabled relatively quick and accurate ocean surface reconstruction (Bergamasco et al. [1]). This included the open-source release of a Wave Acquisition Stereo System (WASS), intended to streamline the process of surface reconstruction over large areas (30 m × 30 m) and facilitate further research. WASS relies on the Lambertian assumption of a diffusely lit surface and a surface-parallel viewing axis to deal with specular reflection. Although the Lambertian assumption and required camera setup is quite limiting, under favorable conditions, WASS has been shown to provide reliable results. Zavadsky et al. [2] applied WASS to measure water surface displacements in a 0.20 m × 0.40 m area inside a wave flume. The authors noted greater noise in the measurements than those recorded by a typical surface-piercing wave probe. The authors also indicated difficulty in achieving proper illumination of the water surface.

Several other stereo image methods that bear some resemblance to the techniques used in particle image velocimetry (PIV) have emerged for laboratory applications. Wang et al. [3] measured water surface deformations caused by solitary waves using stereo image reconstruction. A high-powered overhead light was used to illuminate a dot-matrix on the surface, formed by high-expansion polyethylene granules. Two onlooking cameras were used to create stereo image pairs for surface reconstruction. The measurements showed good agreement with the surface-piercing probes placed in the study region. The authors note, however, that the method was not suited for large surface deformations that might cause parts of the dot-matrix to fold onto itself or overlap. Gomit et al. [4] measured the evolution of surface displacements in the wake of a ship, using two cameras located above the water surface as a stereo pair and a third below the water surface for reference images. The water was seeded with a standard PIV tracer and a submerged high-powered laser was used to beam a grid of light through the water column. Due to the use of high-powered laser sheets, the method was not easily adaptable to larger study areas (on the order of meters).

Single-camera setups are used to determine the dynamic three-dimensional topography of a water surface by exploiting the properties of light refraction (e.g., Moisy et al. [5]; Cobelli et al. [6]; Gomit et al. [7]). These methods typically involve the projection of a pattern from a light source through the water surface. Observed distortions are used to determine the surface topology by calculating the displacement field with respect to the reference image, as it appears in calm water conditions. This technique relies on a transparent water column with no discontinuities in order to maintain a clear line of sight to the projected image. As such, refraction-based methods are generally limited to tests considering non-breaking wave conditions.

A number of other studies have used edge-detection methods to locate the water surface in images captured by a single side-looking camera (e.g., Erikson and Hanson [8]; Du et al. [9]; Viriyakijja and Chinnarasri [10], Foti et al. [11]). One of the main challenges for edge-detection methods is achieving strong and consistent image contrast to facilitate the delineation of material boundaries. Image noise and inconsistent light intensity gradients across the water surface can cause conventional edge-detection methods to return undesirable broken boundary lines or false edges.

Active contour models, known as snakes, are popular machine vision techniques used to determine the outline of an object captured in an image. Duncan et al. [12] used a classic snake model to delineate surface profiles of the spilling breakers. However, the relatively poor convergence of snakes on concave boundaries and their stringent initialization requirements (Xu and Prince [13]) has generally prevented its application to studying surfaces with complex deformations.

In this paper, an image-based method for the automatic detection of water surface levels using color segmentation is presented. A demonstration of the algorithm is given by considering three regular wave conditions producing variable breaker types on the face of a model breakwater. The results were generated using images taken from a single side-looking camera. The tests were conducted in the standard ambient lighting conditions present in the laboratory; no special treatment was required. The method presented entails a relatively simple setup procedure and offers an inexpensive means of performing detailed spatial measurements of water surface profiles in wave flume tests.

## 2. Experimental Model

### 2.1. Model Configuration

As part of a collaborative research project undertaken by the University of Ottawa, and Baird and Associates, a series of experiments were conducted in a 64-m long × 1.22 m wide × 1.22 m tall wave flume at the National Research Council of Canada's Ocean, Coastal, and River Engineering (NRC–OCRE) laboratory in Ottawa. The tests were designed to investigate the hydrodynamic loading on individual armor units, in order to better understand the physical mechanisms leading to armor layer failure. A simple sloping bathymetry was built by filling a series of parallel templates, running length-wise in the flume, with crushed stone, before finishing the surface with grout. The breakwater cross-section was located at the top of the 1:40 foreshore slope, with the structure's toe at an elevation of $z = 0.22$ m above the flume bottom. The series of tests considered for the current work were conducted using a simplified breakwater cross-section, whereby the armor layer was placed on top of a solid impermeable surface set at a 3:4 incline. The impermeable slope consisted of a single, continuous, sheet of polyvinyl chloride (PVC), fixed to a rigid stainless steel frame for support. Individual Core-Loc armor units were placed onto the PVC using a grid painted on the upper surface of slope, as a guide. Care was taken to ensure that the placement guidelines, prescribed by the unit manufacturer (Concrete Layer Innovations, CLI), were satisfied. Upon completion of the armor layer, a rigid waterproof adhesive was applied in small beads at points of contact between adjacent units and between each unit and the impermeable PVC slope. These measures were taken to ensure that the armor layer remained stable during the wave tests as only the hydrodynamics within and above the armor layer were of interest at this stage of the work. A photograph of the model structure is provided in Figure 1b.

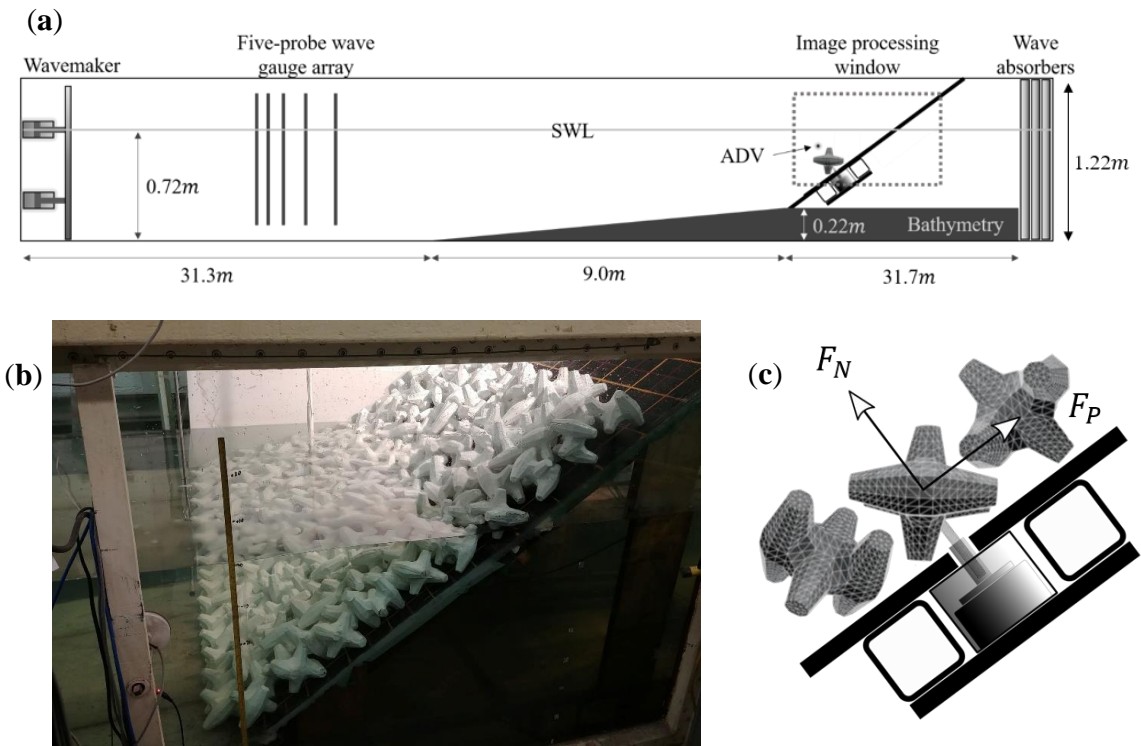

**Figure 1.** (**a**) Diagram of the model setup (not to scale); (**b**) photograph of the model structure; and (**c**) schematic of the armor unit-force sensor coupling mechanism.

## 2.2. Instrumentation

Various devices were deployed to measure water surface elevation, velocity, and the forces and moments acting on individual armor units embedded within the armor layer. A five-probe array of capacitance-type wave gauges (Akamina AWP-24-2) was installed offshore of the toe of the bathymetry. Recorded time-series of water surface elevation from three of the probes were processed using the algorithm outlined by Mansard and Funke [14], to obtain the incident and reflected wave components. A single capacitance-type wave gauge was used to record wave conditions that were slightly seaward of the structure toe, as depicted in Figure 1a. All wave probes were configured to sample at 50 Hz.

A six degrees-of-freedom (6 DoF) force sensor (ATI Mini45) was used to measure the wave-induced forces exerted on an individual armor unit located on the slope, at an elevation approximately coincident with the location of the maximum rundown. The force sensor was housed within a watertight compartment that was rigidly mounted to the stainless steel frame on the underside of the slope, as shown in Figure 1c. Special consideration was given to the instrumented unit to ensure that it did not touch any adjacent armor units or the rigid PVC slope below. The placement guidelines that were observed elsewhere were not strictly enforced for the instrumented armor unit, as doing so would have interfered with the force measurement. Rather, where contact would have been made, a small gap was maintained to ensure that the recorded forces were solely induced by hydrodynamic flows caused by waves breaking on the slope. The data acquisition system was configured to sample the analog force signal at 1000 Hz.

A Nikon D5100 camera, equipped with a Nikon 50 mm f/1.8G AF-S NIKKOR FX lens, was mounted to a wall running parallel to the flume, with approximately 2 m clearance. Images were recorded at 60 frames per second (FPS), with a resolution of 1920 × 1080 pixels. The corresponding field of view (FOV) was approximately 2 m × 1 m, with each pixel representing a, roughly, 1 mm × 1 mm area on the plane of interest. The camera was positioned such that the optical axis passed through the center of the region of interest, as shown in Figure 2. After aligning the camera such that the image plane was parallel to the glass-viewing window, it was fixed in place to prevent any undesired changes to its

spatial orientation. Four control points were marked within the camera's field of view, providing a simple means to check for possible changes to orientation, prior to any post-processing of the images. The camera was triggered remotely by a script running on a dedicated laptop, recording the start time (synced to atomic clock via Network Time Protocol; Mills, [15]) at the beginning of each test. The recorded start time was later used to synchronize the video with the measurements recorded by the data acquisition system.

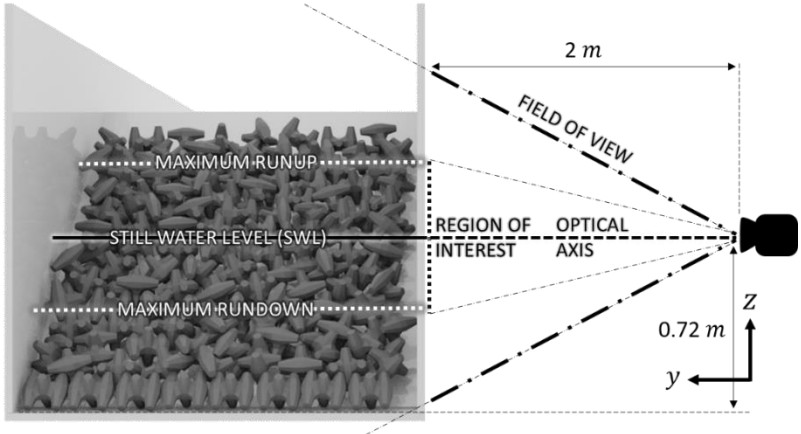

**Figure 2.** Diagram of the camera setup.

A side-looking ADV (Nortek Vectrino) sampling at 1000 Hz was installed a small distance in the slope-normal direction from the instrumented armor unit (shown in Figure 1a) for measuring the X- and Z-direction (see Figure 2) fluid velocity, above the armor layer during the tests.

### 2.3. Test Conditions

Before constructing the model breakwater cross-section, a series of tests were conducted to calibrate drive (command) signals for the wave generator. For this set of tests, a single wave probe was positioned at a location that would later coincide with the structure's toe. Variations of different drive signals were tested until the measured wave periods and wave heights at the aforementioned wave probe matched the target values. Target values for the wave period were chosen to produce a wide range of conditions, including various breaking wave types and extreme runup and rundown elevations. Characteristics for the regular waves considered in the present work are summarized in Table 1.

**Table 1.** Calibrated regular wave conditions for the image processing tests.

| Test Name | Wave Height (m) | Wave Period (s) | Breaker Type |
|:---:|:---:|:---:|:---:|
| **H2T1** | 0.20 | 1.4 | Plunging |
| **H2T2** | 0.20 | 2.0 | Collapsing |
| **H2T3** | 0.20 | 2.4 | Surging |

## 3. Image Processing

### 3.1. Camera Calibration

Lens distortion is a common effect observed in images produced by a typical camera and should be corrected for. Camera calibration is a term often used to describe the process through which the camera parameters needed to correct for lens distortion are obtained. This is typically performed by modeling the image-taking system as an idealized pinhole camera. An ideal pinhole camera consists of a small aperture with no lens, through which light passes to create an image of the viewed scene.

Conceptually, this process can be understood as the projection of 3D points in the world coordinate space to a 2D image plane, and can be expressed as:

$$sm' = A[R|t]M' \tag{1}$$

or, alternatively,

$$s \begin{bmatrix} u \\ v \\ 1 \end{bmatrix} = \begin{bmatrix} f_x & 0 & c_x \\ 0 & f_y & c_y \\ 0 & 0 & 1 \end{bmatrix} \begin{bmatrix} r_{11} & r_{12} & r_{13} & t_x \\ r_{21} & r_{22} & r_{23} & t_y \\ r_{31} & r_{32} & r_{33} & t_z \end{bmatrix} \begin{bmatrix} X_w \\ Y_w \\ Z_w \\ 1 \end{bmatrix} \tag{2}$$

where $s$ is the scale factor; $m' = \begin{bmatrix} u & v & 1 \end{bmatrix}^T$ is the 2D image coordinate vector, expressed in pixels; and $M' = [X_w\,Y_w\,Z_w\,1]^T$ is the 3D coordinate vector of a point in the world coordinate space.

The joint rotation-translation matrix, $[R|t]$, is responsible for the transformation of a world point, $M'$, to a coordinate system fixed with respect to the camera. This transformation is dependent on information contained within the viewed scene and is, thus, often referred to as the extrinsic camera parameter matrix.

The intrinsic camera parameter matrix, $A$, is responsible for transforming the points in the camera coordinate space to the image coordinates. The intrinsic camera parameter matrix consists of: the focal lengths along the x- and y-axis, $f_x$ and $f_y$, and; the coordinates of the optical center, $c_x$ and $c_y$, expressed in pixels, typically equal to the coordinates of the image center. Note that for pixels with an aspect ratio of $\alpha = 1$ (i.e., square pixels), $f_x = f_y$.

The intrinsic and extrinsic camera parameter matrices described above were calculated for the camera setup used in the present study, by following the general procedure outlined by Zhang [16]. In theory, the camera parameter matrices $A$ and $[R|t]$ can be estimated from three images taken by the camera, each containing a unique view of an object (typically some variation of a repeated pattern), with known geometry and easily identifiable features. In practice, however, it is recommended to take, at minimum, ten unique images, in order to offset the effect of noise and other sources of error. In the present work, the authors opted for 50 images taken of a $9 \times 6$ checkerboard pattern, in various positions within the field of view. These images were converted to grayscale and passed to a function available in the OpenCV library (Bradski, [17]), for further processing. In this function, the process of estimating $M'$ is automated through the application of edge-detection and Hough Line Transforms, to find the position and orientation of the pattern within the field of view. Using the known image coordinates of corners within the calibration pattern, $m'$, and their corresponding estimated world coordinates, $M'$, the intrinsic and extrinsic camera parameters were computed.

*3.2. Distortion Correction*

The distortion inevitably caused by the lens in a typical camera could be classified into two types—radial and tangential distortion. Radial distortion is attributed to light bending more near the edges of the lens, compared to light passing through the center. Radial distortion is responsible for the familiar 'fisheye' effect often apparent in images where the distortion becomes more severe with distance from the image center. Tangential distortion occurs when elements of the lens are misaligned, causing a decentering of the images that are produced.

Once the intrinsic and extrinsic camera parameters (defined in Section 3.1) are known, the radial and tangential distortion (first and second-term on the right-hand-side of Equation (3) respectively) can be corrected by remapping the image coordinates according to,

$$U = \begin{bmatrix} u \\ v \end{bmatrix} = \left(1 + k_1 r^2 + k_2 r^4 + k_3 r^6\right) U_d + dU \tag{3}$$

where $U_d = \begin{bmatrix} u_d & v_d \end{bmatrix}^T$ is the image coordinate vector of the distorted image; $k_1$, $k_2$, and $k_3$ are the radial distortion coefficients; $r = \sqrt{(u_d - c_u)^2 \, (v_d - c_v)^2}$ is the distance between point $U_d$, and the image center; and $dU$ is the tangential distortion correction, given by

$$dU = \begin{bmatrix} 2p_1 u_d v_d + p_2(r^2 + 2u_d^2) \\ p_1(r^2 + 2v_d^2) + 2p_2 u_d v_d \end{bmatrix} \tag{4}$$

where $p_1$ and $p_2$ are the tangential distortion coefficients.

The distortion coefficients were calculated by the method of least squares, using OpenCV (Bradski, [17]), minimizing the error between the points detected in the image and the projection of their real locations (in 3D world-space) to the image plane via Equations (1) and (3) Once the distortion coefficients were obtained, Equation (3) was used to remap the pixels of each frame from the video footage recorded during the tests.

### 3.3. Surface Tracking Algorithm

The problem of measuring water surface elevation using an imaging device can be understood, more generally, as a classification problem. To solve this problem, the authors of the current study developed a method to classify each pixel in an image as belonging to one of two classes:

Class 1: pixel ∈ water below the surface (i.e., pixel belongs to water below the surface), or

Class 2: pixel ∉ water below the surface (i.e., pixel does not belong to water below the surface).

In this section, the process through which the pixels in an image were classified according to the scheme outlined above is described.

Following the corrections applied for lens distortion, described in Section 3.1, the images were cropped to the region of interest. In the case presented herein, an inclined line can be drawn to approximate the upper boundary of the armor layer, shown as a white dashed line in Figure 3a. Since extracting information from within the internal porous structure of the armor layer is beyond the scope of this study, anything below the white dashed line is considered superfluous, necessary only to maintain the square shape requirement of the image arrays. In this sense, the white dashed line can also be thought of as an image-processing boundary, and will be referred to as such herein.

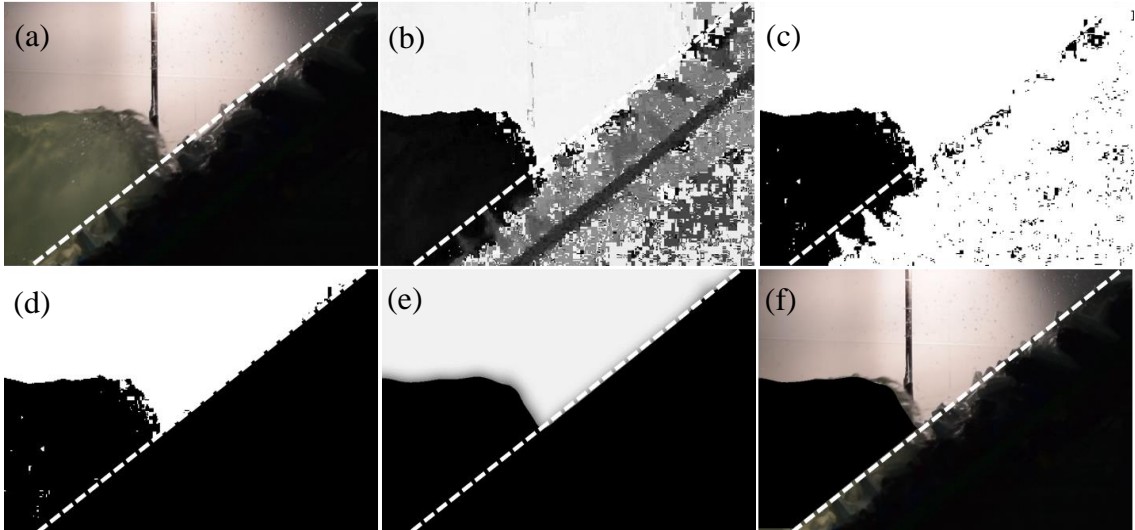

**Figure 3.** Visual representation of the image-processing algorithm: (**a**) Original RGB image; (**b**) conversion to hue; (**c**) threshold on hue; (**d**) boundary insertion; (**e**) Gaussian blur; and (**f**) boundary removal.

Segmenting the image into the classes described above begins with a color space transformation of the source image from RGB (Red/Green/Blue) to HSV (Hue/Saturation/Value) format. The result of this transformation is then split into its three component channels to isolate the hue. It is immediately clear from the grayscale representation of hue (shown in Figure 3b) that the hue of the water is relatively consistent and, more importantly, unique amongst the other components of the image. By analyzing a few of these images, lower and upper bounds for the hue that described the range for class 1 pixels, under the conditions for this study, were determined. Using these lower and upper bounds, a simple threshold operation was performed, yielding the binary image shown in Figure 3c. At this stage, the black pixels in Figure 3c could be thought of as potential class 1 candidates. For the purposes of this study, it was desirable to attain a continuous representation of the two classes with relatively smooth boundaries separating them. An effective means to achieve this was found by reducing high-frequency noise in the images by applying a Gaussian filter. While a kernel size of $15 \times 15$ was found to yield the best results, the amount of surface detail could be varied by modifying the size of the kernel. Application of the Gaussian filter tended to soften the gradients, removing small details while retaining the overall shape. Before applying the Gaussian filter, the authors found it advantageous to temporarily turn regions of the image that coincide with objects or surfaces that the water interacts with to class 1 pixels. Neglecting this step would often lead to the erosion of class 1 regions that spanned a distance of less than or equal to the kernel size. This effect subsequently caused slight underestimation of the maximum runup elevation and of the water depth on the structure, as it approached the kernel size. In order to avoid this, pixels at an elevation equal to or lower than the image-processing boundary were temporarily converted to class 1, as depicted in Figure 3d. It should be noted, however, that larger kernel sizes tended to cause stronger melding of regions with similar values of hue. Since the authors found it necessary to temporarily change the hue of the structure to match that of the water, one might expect larger errors in the computed surface levels in the region (approximately, one kernel wide) between the water and the structure. Although the effect was relatively small in the present study, it might be of larger concern for applications requiring stronger smoothing. A Gaussian filter was then applied to the image, shown in Figure 3e, removing the undesirable high-frequency noise while retaining the overall shape. Finally, a simple threshold was performed, using the half-way point between white (255) and black (0), and the temporary conversion of pixels under the image-processing boundary was reverted. The result of the final operation was superimposed over the source image; shown in Figure 3f.

### 3.4. Coordinate Transformation

The output of the surface-tracking algorithm described above (Section 3.1) was used as the basis for all subsequent measurements and calculations. Typically, the information extracted from the final processed image is a set of image coordinates corresponding to the points of interest. In this study, the physical coordinates of the tracked features occur within a single plane, perpendicular to the optical axis. Thus, a simple relationship between the observed image coordinates and their corresponding physical coordinates could be derived. This transformation could be expressed as:

$$\vec{X} = \left[ \begin{array}{c} (u - u_0)x_p \\ (v_0 - v)z_p \end{array} \right] \tag{5}$$

where $u$ and $v$ are the image coordinates of a point of interest and $u_0$ and $v_0$ are the image coordinates of the coordinate system origin. The physical distance spanned by a single pixel in the horizontal and vertical directions ($x_p$ and $z_p$, respectively) varied with distance along the optical axis. In the current work, the values for $x_p$ and $z_p$ were derived by taking an image of a $6 \times 6$ cm grid fixed to the glass sidewall of the flume (as seen in Figure 4), correcting the image for lens distortion and dividing the grid resolution by the average number of pixels spanning each cell.

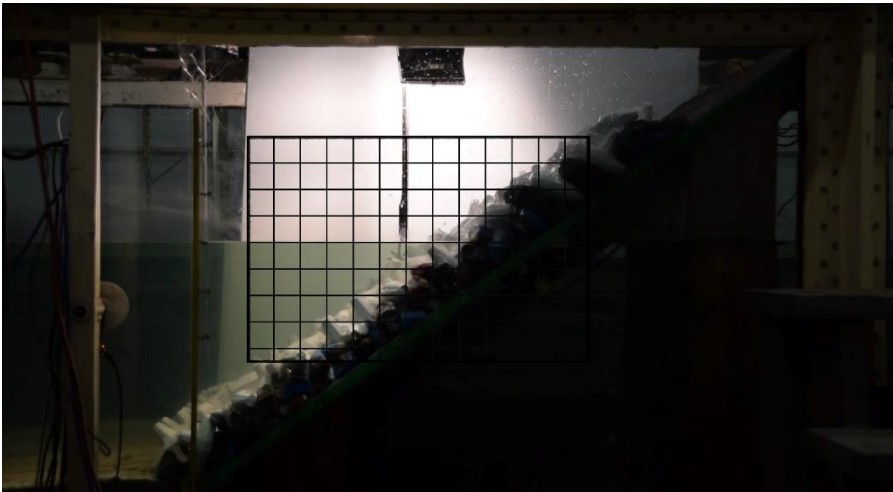

**Figure 4.** Distortion-corrected image (full field of view) with the coordinate transformation calibration grid superimposed over the region of interest.

## 4. Analysis and Discussion of Results

### 4.1. Validation of the Image-Processing Algorithm

Validation of the image-processing algorithm was performed by comparing a series of computed and visually delineated water surface profiles at various times during the wave cycle. It was noted that some degree of subjectivity was involved in performing this comparison. For the majority of the total length of visually delineated water surface profile, it was possible to identify the air–water interface to within an accuracy of ±2 pixels. This level of accuracy corresponded to, roughly, ±1% error (in terms of the incident wave height). There were, however, isolated segments of the water surface at particular stages of the wave cycle where the boundary between the air and water was less clear. The greatest difficulty in discerning the water surface was encountered for the images taken shortly after the time of maximum rundown, when the incoming wave was, either in the process of breaking or had already broken, before proceeding to run up the structure. In these cases, the relatively high amount of turbulence and entrained air produced a blurred region (spanning, perhaps, up to 20 pixels in the vertical direction) within the vicinity of where one would reasonably expect the water surface to be. It was only during these short portions of the wave cycle that delineating the water surface required a higher degree of subjectivity. It is worth noting that a capacitance-type wave gauge measuring at the same location would experience similar difficulty, as such sensors are known to have trouble discerning the interfaces between solid green water, aerated white water, and air. Altogether, the authors estimate that the error in the position of the visually delineated water surface profiles presented in this section was roughly on par with a high-precision capacitance-type wave gauge.

Two frames, taken from the videos recorded during the tests with the shortest ($T$ = 1.4 s) and longest wave periods ($T$ = 2.4 s), were selected for the comparisons presented in Figure 5a,b and Figure 5c,d, respectively. These images were selected to be representative of the full range of conditions for the tests considered in the present work. In Figure 5, the coordinate system was normalized by the characteristic length of the armor units (i.e., $c$ = 0.12 m) and was defined with respect to the horizontal coordinate of the structure's toe ($x$ = 0 m) and the still water level ($z$ = 0 m).

Figure 5a shows an incident wave in the process of breaking on the structure, creating conditions similar to those that made visually delineating the surface around the location of minimum rundown more uncertain. This was, in general, the case for the segment extending from $X$ = 0.36 m, up to a point within the armor layer, where the surface fades from sight entirely (approximately, $X$ = 0.54 m). The relatively large difference between the computed and visually-delineated water surface elevation observed in this region was likely due to the different biases inherent in the algorithm's and the authors'

decision-making process. When probed to make a judgement on the location of the water surface when there exists a clear boundary between the air and water, there is a tendency to give similar answers. The differences in bias only become apparent when asked to make a judgement where no clear boundary exists. Under such conditions, it is not possible to quantify the accuracy, since there is no objective correct answer.

In Figure 5b, the computed and visually delineated surface profiles are compared during rundown, when the free surface was roughly half-way between the maximum runup and the maximum rundown. Here, the difference between the two profiles was relatively small, with the greatest difference occurring at the location where the surface intersects the image-processing boundary. During this stage of the wave cycle, there was little difficulty in visually delineating the surface, and so the small differences in free-surface position were likely due to the chosen threshold values for hue (see step (c) in Figure 3). It was observed that as the depth of water on the structure approached zero, the hue of the water and that of the background became increasingly indistinguishable. This effect seemed to be more pronounced at certain locations along the surface of the structure. This dependence on location led the authors to believe that the effect might be related to the angle of perspective, aligning with the path of ambient light refracted through the surface.

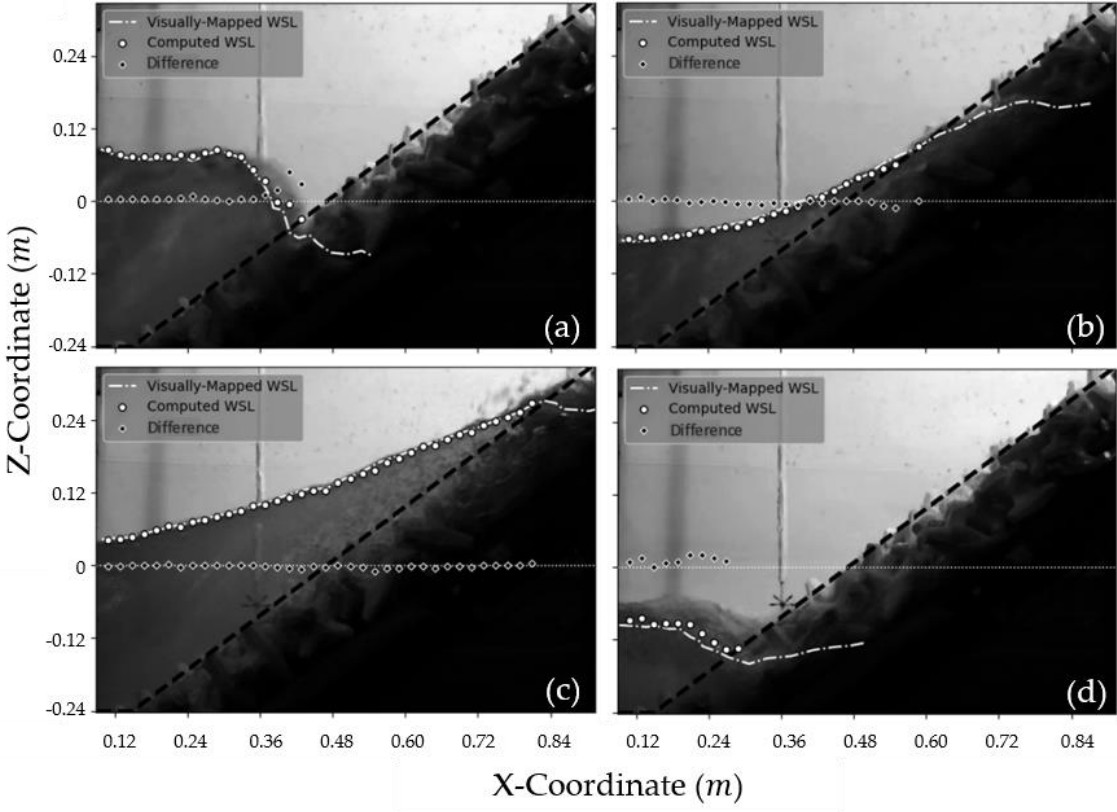

**Figure 5.** Comparison of the computed and visually delineated water surface profiles at different stages of the wave cycle for test H2T1: (**a**) Breaker arriving at the structure, (**b**) mid rundown, (**c**) maximum runup, and (**d**) maximum rundown.

For the maximum runup event, depicted in Figure 5c, relatively good agreement was observed between the computed and visually delineated surface profiles, over their entire length.

A comparison of the computed and visually delineated surface profiles for the maximum rundown event is given in Figure 5d. At this point in time, the water rushing out of the armor layer merges with water above the armor layer, at the location of maximum rundown. This caused a turbulent breakup of the otherwise smooth surface, causing a significant reduction in the amount of reflected light arriving at the camera, from the surface. This reduction in reflected light caused the hue of the surface water to

shift toward the range that it relied upon, for the threshold (see step (c) in Figure 3). Typically, this effect became more pronounced for wave conditions that produce larger maximum rundown levels. Figure 5d shows the effect of this in the most extreme case considered in the present work. The average normalized difference between the computed and visually delineated surface profiles reached a value of roughly 0.08, corresponding to a 5% difference, relative to the incident wave height.

### 4.2. Comparison of Point Velocity Measurements and Computed Runup Velocity

With a few additional computations, it was possible to estimate the time-series of runup velocity, just above the surface of the structure. In order to perform this calculation, a line of pixels, running parallel to the surface of the structure was isolated from the main image, and the location of the water surface along this line of pixels was determined. The location of the water surface along this line (referred to hereafter as a tracked point) was stored on a frame-by-frame basis and was used to compute the runup velocity according to,

$$R_w = \left[ \begin{array}{c} \Delta u * x_p * f_s \\ \Delta v * z_p * f_s \end{array} \right] \tag{6}$$

where $f_s$ is the sample frequency, equivalent to the FPS output of the camera.

At times when the slope of the water surface was close to that of the structure itself, erratic fluctuations in the location of the tracked point were often observed. This tended to produce undesired fluctuations in the computed runup velocity. In order to counter this effect, the average change in location of multiple tracked points were used for the calculation presented in Equation (6) A typical arrangement of these tracked points, at different stages during the wave cycle, is given in Figure 6a–d. For visual clarity, the spacing between the inclined lines was increased and the total number of tracked points was decreased, relative to the values used for the automated computation by the algorithm. The authors found that using a total of 30–50 tracked points along the lines separated by equal intervals of 1–2 pixels, yielded the best results.

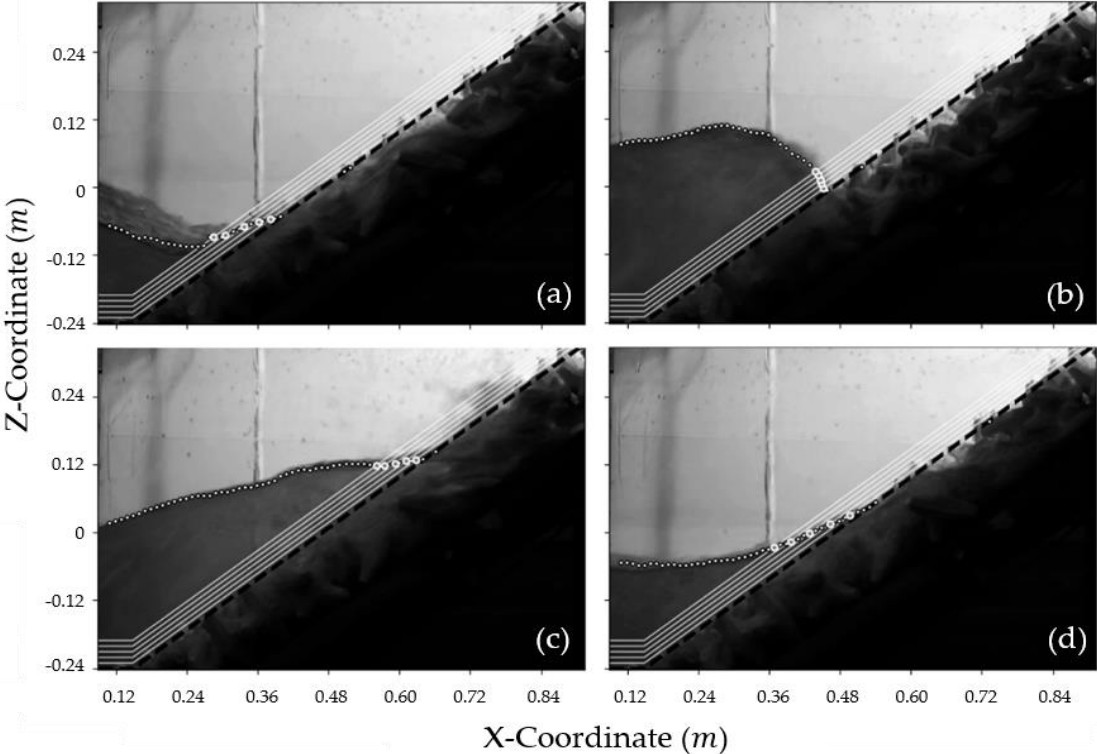

**Figure 6.** Visual demonstration of the tracking point acquisition used for the runup velocity computation at (**a**) maximum rundown; (**b**) mid runup; (**c**) max runup; and (**d**) mid rundown.

In Figure 7, the computed runup velocities were compared to the ADV measurements over the first five stable wave cycles for the test conditions presented in Table 1. Due to its vertical location, the ADV became exposed to the air for a short period of time, during each wave cycle, when the water surface elevation on the structure was near minimum. Portions of the signal corresponding to these periods were removed for clarity. It is important to note that the comparisons shown in Figure 7 were made between the Lagrangian (runup velocity) and Eulerian (ADV measurements) descriptions of velocity, and, as such, must be interpreted with caution. There were, however, special circumstances under which it could be reasoned that the runup velocity and ADV measurements were, essentially, two estimates of the same quantity (the only difference being the means through which they were obtained). These special circumstances occur twice per wave cycle, each occurring when the free surface passed the location of the ADV—once during runup and, again, during rundown. At these two instances, the average location of the tracked points used to compute the runup velocity roughly coincided with the location of the ADV. Occurrences of these special circumstances are indicated by the dark grey diamonds in Figure 7.

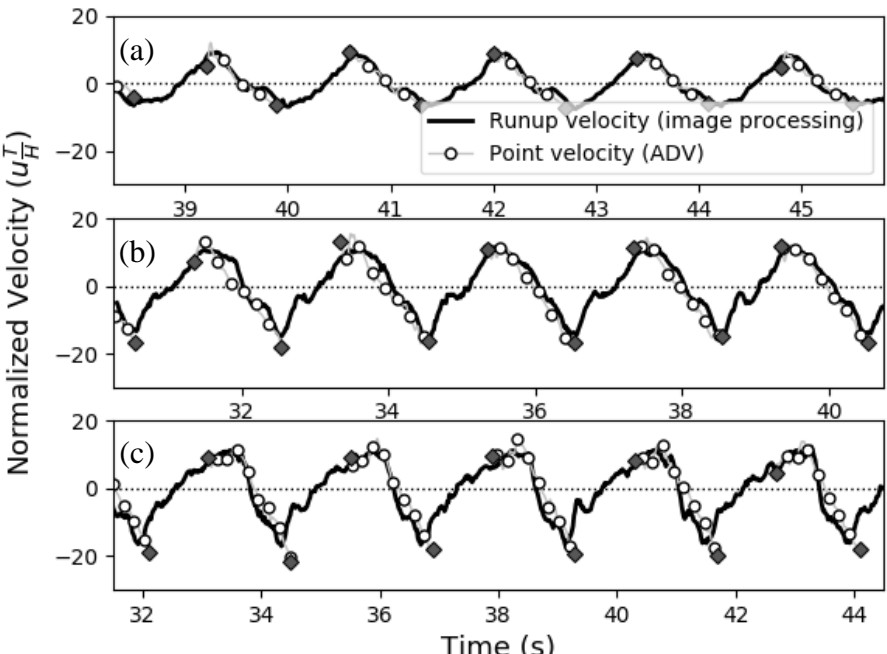

**Figure 7.** Comparison of the computed runup velocity and the ADV point velocity measurement time-series for the wave periods (**a**) $T = 1.4$ s, (**b**) $T = 2.0$ s, and (**c**) $T = 2.4$ s.

For all three wave periods, relatively good agreement was observed between the computed and the measured velocities, when the free surface arrived at the location of the ADV during runup. A similar level of agreement was observed at the time the free surface passed below the ADV, during the rundown for the two shorter wave periods, as shown in Figure 7a,b, respectively. For the largest wave period, shown in Figure 7c, the computed velocity was slightly underestimated, relative to the measured value, at the times when the free surface passed below the ADV. A close examination of the rundown velocities for all three wave periods, revealed a trend that could be explained by referring to an observation noted in an earlier section. In Section 1, the authors noted an increasing tendency for the algorithm to overestimate the water level at maximum rundown, as the wave period increased. This tendency was attributed to the turbulent breakup of the surface during rundown, causing a shift in the hue of the surface water into the range that was relied upon for the threshold operation. While this effect was negligible for the shortest wave period ($T = 1.4$ s), it grew more pronounced for the longer wave periods, causing up to 5% error (relative to the incident wave height) for the largest wave period considered ($T = 2.4$ s). Owing to the algorithm's reliance on the change in water level to compute

runup velocity, any error in the determination of the water surface position was inevitably carried over into the computation of the runup velocity.

*4.3. Buoyancy Correction*

For coastal structures in general, it is expected that some of their components will be fully immersed in water during some parts of the wave cycle and completely exposed to air during others. This poses an additional challenge when measuring wave-induced forces on such components. Due to the sharp discontinuity in fluid density across the air–water interface, the buoyant force exerted on an object as it passes through the interface will vary in magnitude. If one were to record the time-history of force acting on this object, the measured signal would include hydrodynamic (drag and inertial) and hydrostatic (buoyant) force components. For an object remaining submerged, this does not pose a problem, since the force-measuring device can be zeroed while the object was submerged, thereby pre-emptively removing the buoyant force component from the signal. Otherwise, the buoyant force can be easily computed and removed from the measured force signal in post-processing. For an object passing through the interface, however, additional information is needed to remove the buoyant force from the measured force signal; the relative positon of the interface with respect to the object and knowledge of how the submerged volume of the object changed, as it passes through the interface. In the current work, the image-processing algorithm was extended to handle this problem by computing a time-varying buoyancy-correction and applying it to the measured force signal.

The forces presented in this section were measured on an armor unit located at an elevation approximately coincident with the maximum rundown elevation. The position of the armor unit on the structure slope was such that it was fully submerged in still water before the start of each test, remained submerged during most of the wave cycle, but became partially emergent for a period of time around maximum rundown. The fully submerged buoyant force was pre-emptively removed from the measured force signal by re-zeroing the force sensor in still water conditions (i.e., when the armor unit was fully submerged), prior to the start of each test. Thus, the appropriate buoyant force correction applied to the measured slope-parallel and slope-normal force signal is given by the piecewise linear function

$$
F_B^C = [\sin\alpha \ \cos\alpha] \begin{cases} 0, & Z_{WSE} \geq Z_{UB} \\ (V_{Total} - V_{Sub})\rho_w g, & Z_{LB} < Z_{WSE} < Z_{UB} \\ V_{Total}\rho_w g, & Z_{WSE} \leq Z_{LB} \end{cases} \tag{7}
$$

where $\alpha$ is the slope of the structure with respect to the horizontal, $Z_{WSE}$ is the average water surface elevation surrounding the armor unit, $Z_{UB}$ and $Z_{LB}$ are the elevations of the upper and lower boundary of the armor unit, $V_{Total}$ is the total volume of the armor unit, $V_{Sub}$ is the submerged volume of the armor unit, $\rho_w$ is the density of water, and $g$ is the acceleration due to gravity.

Ideally, the value for the water surface elevation, $Z_{WSE}$, appearing in Equation (7) would be taken as the average water surface elevation contacting the armor unit. However, the visual information required to infer the water surface levels immediately surrounding the armor unit was hidden from the field of view. Instead, an approximation for $Z_{WSE}$ was computed for each video frame as the elevation at which the water surface intersected the image-processing boundary. An illustration of this approximation is provided in Figure 8a,b, corresponding to the maximum runup and maximum rundown events, respectively.

Under conditions such as those presented in Figure 8b, when the water surface elevation intersected the armor unit (i.e., $Z_{LB} < Z_{WSE} < Z_{UB}$), the submerged volume, $V_{Sub}$, was computed using various tools available in the open source Trimesh library (Dawson-Haggerty [18]). First, a horizontal plane with an elevation equal to $Z_{WSE}$ was used to slice a georeferenced 3D mesh of the armor unit, leaving only parts of the mesh that exist below $Z_{WSE}$. Next, the open face of the sliced mesh was triangulated in order to close the volume, satisfying the watertight condition required for the volume computation.

Finally, the volume of the submerged portion of the mesh was computed. Details on the computation procedure can be found in Zhang and Chen [19].

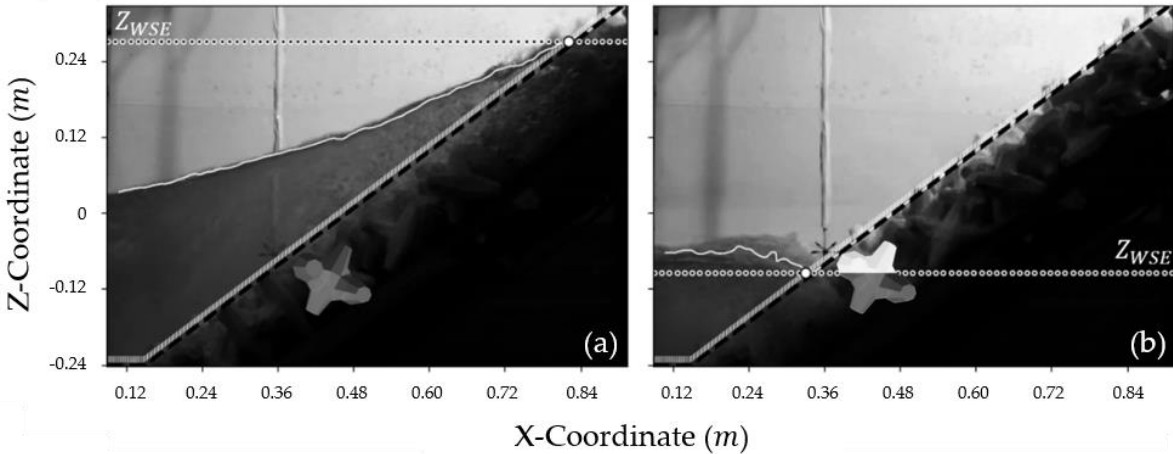

**Figure 8.** Image-processed runup elevation ($Z_{WSE}$) used for computing buoyancy correction at times corresponding to (**a**) maximum runup and (**b**) maximum rundown for test H2T2.

In Figure 9, time-series of the measured and buoyancy-corrected forces acting on the armor unit are shown for the first five stable waves arriving at the structure with a wave period $T = 2.0$ s. Since the force sensor was re-zeroed when the instrumented armor unit was fully submerged, the buoyancy correction term increased from zero when parts of the unit became exposed to the air. Furthermore, the buoyancy correction term continued to increase until maximum rundown was observed, at which point the exposed volume of the instrumented armor unit was at a maximum.

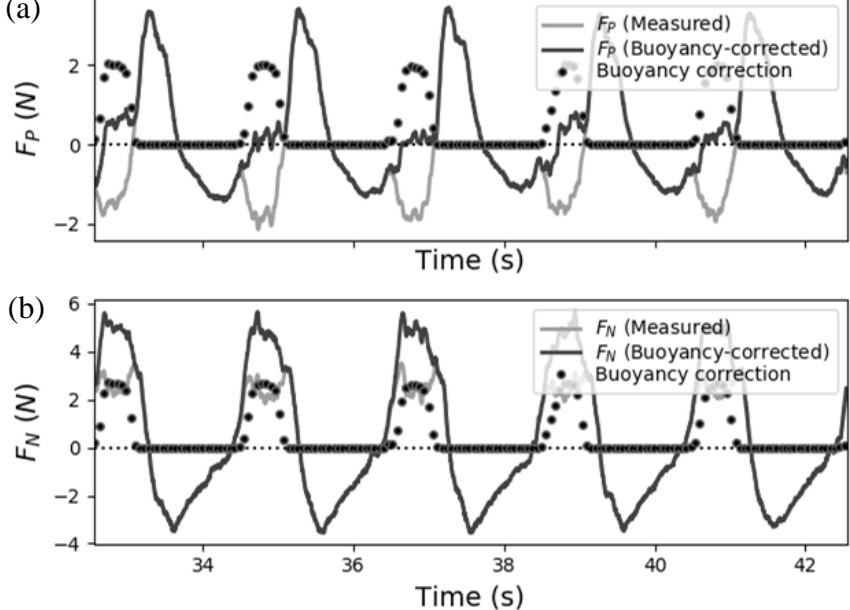

**Figure 9.** Time-series of measured and buoyancy-corrected forces acting on the armor unit in the (**a**) slope-parallel and (**b**) slope-normal direction for test H2T2.

From the slope-parallel forces shown in Figure 9a, it was noted that the minimum of the measured force was, approximately, coincident with the occurrence of maximum rundown. For these particular test conditions (notably, armor unit location and wave characteristics), applying the buoyancy correction to the measured force signal caused a shift in the time at which the minimum slope-parallel force was observed. In fact, since the buoyancy-corrected force signal oscillated around $F_P = 0$ N at the time when

the minimum measured force was observed, it could be inferred that the measured slope-parallel force at this time was largely (if not entirely) caused by the increase in the apparent weight of the armor unit. The minimum buoyancy-corrected force appeared to occur just before the armor unit started to become exposed to the air (i.e., when the buoyancy correction term began to increase from zero). This suggests that the decreasing area of the armor unit exposed to the flow had a much greater influence on the hydrodynamic forces exerted on the unit than any subsequent changes to the acceleration or velocity of the rundown.

It can be seen in Figure 9b that the maximum slope-normal forces occurred around the time of the maximum rundown (i.e., when the buoyancy correction term was at a maximum). Furthermore, the peaks of the slope-normal force were observed to occur when the slope-parallel force oscillated around $F_P = 0 \ N$, suggesting that the maximum slope-normal forces were dominated by drag and inertia, as opposed to lift.

## 5. Conclusions

In this study, the authors present a detailed description of an innovative image-processing method developed to track water surface elevation under a wide range of wave conditions. The image processing method was further extended to compute wave runup characteristics on a rubble mound structure. Using the computed wave runup characteristics, the time-varying buoyant force acting on an armor unit near the location of maximum rundown was computed and applied to correct the measured force signals. Several conclusions were drawn from the results; these are summarized as follows:

1)  Relatively good agreement was observed in the comparisons of the visually delineated and image-processed water surface profiles. Of particular interest, the tracking algorithm demonstrated relatively good accuracy in the conditions not suitable for the application of edge-detection methods.
2)  Some issues in accurately resolving the water surface profile were encountered for the wave conditions that produced relatively large rundown levels on the structure. This was attributed to significant reductions in the light reflected from the water surface due to turbulent mixing of the rundown, with the main body of water immediately seaward of the structure.
3)  Comparisons of the computed and measured runup and rundown velocities (under conditions when such a comparison is appropriate) suggest that the method used for the computation could yield reliable estimates of runup and rundown velocity on a rough surface structure.
4)  The image-processing results were used to correct the measured forces on a breakwater armor unit, for the effects of buoyancy. In doing so, the combined drag and inertia forces were isolated from the hydrostatic forces at every point during the wave cycle, offering detailed insight into how the forces develop over time.

**Author Contributions:** Conceptualization, S.D., A.C., and I.N.; Formal analysis, S.D.; Funding acquisition, A.C. and I.N.; Investigation, S.D.; Methodology, S.D., A.C., and I.N.; Project administration, A.C. and I.N.; Resources, A.C.; Software, S.D.; Supervision, A.C. and I.N.; Validation, S.D.; Visualization, S.D.; Writing—original draft, S.D.; Writing—review & editing, A.C. and I.N. All authors have read and agreed to the published version of the manuscript.

**Funding:** The authors would like to acknowledge the support of the Natural Sciences and Engineering Research Council of Canada (NSERC) through the Collaborative Research and Development Grant (NSERC-CRD) and the NSERC-Discovery Grant, both held by Ioan Nistor, as well as the NSERC PSG-D Doctoral Grant held by Steven Douglas. The financial and scientific support provided by Baird and Associates, Ottawa, is also graciously acknowledged.

**Conflicts of Interest:** The authors declare no conflict of interest.

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
