# Peer review of "Image-Based Measurement of Wave Interactions with Rubble Mound Breakwaters"

_jmse, doi:10.3390/jmse8060472_

Round 1

Reviewer 1 Report

The paper is interesting and introduces an appropriate methodology to analyze hydrodynamic patterns in rubble mound breakwaters. All sections are correctly presented, beginning with a precise analysis of the innovations that the study shows regarding previous references. The “Experimental model” and “Image Processing” parts are very detailed, showing all the main aspects, so there is no need for additional information. The results and conclusions sections are again appropriate. This reviewer finds very interesting the analysis of velocities and buoyancy correction.

Minor revision. Page 15, line 455-456. The authors mention the “agreement with observations documented in a number of past studies (e.g., Sigurdsson, [10]; Sandström, 455 [11]; Juhl and Jensen, [12]; Hald, [13])”. These references should also be included in the Introduction section, with some text of the main aspects. In addition, some more detailed analysis and proofs of this agreement should be presented in the Results section.

Reviewer 2 Report

Review of “Image-based measurement of wave interactions with rubblemound breakwaters”

The authors investigate run-up and run-down. Measurements of the free surface in proximity of the scaled breakwater are performed using a image based technique that exploits different color channel (HSV) to isolate water profile.

Overall the technique presented and results scientifically sound. I find, however, that the authors have presented limited literature on the topic. For example the authors state “most of these experiments, however, were performed in deep water with non-breaking waves”. The paper “An experimental comparison of velocities underneath focussed breaking waves” by Alberello et al., 2018 investigates breaking waves and provide a broad overview of past measurements (see references therein).

At this stage I recommend major revisions. I feel the authors should expand the introductory section and revise previous literature. Other comments are listed below.

  • line 53. The authors report accuracy as 2 pixels (4 mm), but should also indicate the percentage of error over the measurement, I.e. 4 mm is small for waves 4 m high but it is 25% error for waves 2 cm high.
  • In 2.1 the authors specify the flume dimensions, however they only specify the depth in the sketch in figure (0.72m).
  • Table 2.1 the period is in s, if in meters this denotes wavelength
  • The author should specify camera resolution, lens specifications (focal length) and camera parameters (aperture, shutter speed, ISO). I note that a macro lens is usually used for planar measurements and a wide aperture to isolate a narrow in-focus plane. A focal length ~50mm (full frame equivalent) usually reduce image distortion.
  • The author should clearly state what is the field of view (dimensions) and resolution (px/m).
  • Figure 4.1 rather than the dimensionless measurements, actual measurements would be more informative (or dimensionless normalized by wave amplitude) rather than using armor unit. Also “c” is confusing, because normally used to denote wave celerity.
  • Run-up velocity is somewhat misleading. What is actually measured is the displacement of the free surface, in wave theory is important to clarify the frame of reference (Eulerian vs Lagrangian). ADV by contrast measures the velocity in Eulerian frame of reference at a fixed point, comparison is not straightforward, e.g. measurement of the advective velocity component.

Reviewer 3 Report

Please, see the attached document.

Reviewer 4 Report

Journal of Marine Science and Engineering

Image-based measurement of wave interactions with rubblemound breakwaters

by Douglas S., Cornett A. and Nistor I.

Reviewer’s report

The paper presents an innovative image processing method developed to track water surface elevation under a wide range of wave conditions.

The argument is of some interest for the scientific community and can be published after a major revision.

Specific comments:

  1. Introduction - The surface topography of waves has been recovered in the field through 3D photogrammetry by the many works of Benetazzo A. and co-authours, starting from the earlier CENG work (Benetazzo, 2006, CENG). The authours should comment on the advantages/possibility differences and difficulties in applying their approach to situ measurements (e.g. light conditions, reflection, field of view, distances, etc.)
  2. Introduction - In flume applications, one of the advantages of the optical systems is the possibility to have a "continuous" probe within an area. This should be highlighted.
  3. Introduction - Edge detection is only one of the possibilities to given by algorithm of image segmentation. A combination of different image segmentation approaches was used by Musumeci et al. (2013) to track the 3D scour evolution at the toe of a vertical pile.
  4. Introduction - As a matter of fact, there are several works which deal with breaking waves. One interesting application of Vousdoukas et al. (2014) to detect wave-by-wave swash zone processes by the combination of LIDAR data and video images.
  5. Introduction - The Structure from Motion technique and RGB-D images are now widely used in the field of coastal engineering, by using off-the-shelf tools and algorithms (see Petersen et al. 2015; Musumeci et al., 2018). The authors should critically compare such approaches with their own.
  6. Section 2.1 - The use of an impermeable slope differs quite a bit from the actual hydrodynamics of real structures, where there are continuous fluxes within the structure, influencing pressure distribution.
  7. Section 2.2 - How the sampling frequency may hinder results on the impacts of the waves on the structure? Please discuss.
  8. Section 2.2 - L. 133-142 -The measuring approach is interesting; however, it does have limitations. This should be clearly discussed.
  9. Section 2.2 – L. 143-145 - Provide info on the resolution of the images.
  10. Section 2.2 – L. 153 - 1000 Hz? Is this a typo?
  11. Section 2.2 – L. 153 – 155 - I assume that the Vectrino was side-looking. Please specify.
  12. Section 3.3 – L. 244 - The application of the Gaussian filter removes part of foam (air-water mixture) of the roller of the breaking wave. This is the region where most of the breaking generated vorticity is concentrated (Tatlock et al., 2018). Thus, this is the region where the flow interacts more intensively with the structure. How does this affect the results? Moreover, during uprush and downrush the presence of foam may be significant. What is the influence on the runup, rundown estimates?
  13. Figure 3.1 – The letter from second (c) to (e) in the caption are wrong. Please check.
  14. Section 4.1 – L. 279 – The author asses that “it was possible to identify the air-water interface to within an accuracy of ±2 pixels”. Does this hold also in the roller region?
  15. Section 4.1 – Actual comparison with measurements would have been interesting.
  16. Section 4.1 – L. 310 – 311 - This is a bit obscure
  17. Figure 4.1 - Include description of subplots within the caption.
  18. Figure 4.3 - This is quite interesting as an analysis. The authors should provide hints on the engineering implications

References

  1. Musumeci,R.E., Farinella,  M., Foti, E.,  Battiato,  S., Petersen, T.U.,   Sumer, B.M. (2013) Measuring sandy bottom dynamics by exployting depth from stereo video sequences.  Lecture Notes in Computer Science (including subseries Lecture Notes in Artificial Intelligence and Lecture Notes in Bioinformatics) Volume 8156 LNCS, Issue PART 1, 2013, Pages 420-430
  2. Musumeci, R.E., Moltisanti, D., Foti, E., Battiato, S., Farinella, G.M. (2018). 3-D monitoring of rubble mound breakwater damages. Measurement: Journal of the International Measurement Confederation, vol. 117, March 2018, Pages 347-364
  3. Petersen, T.U., Sumer, M., Fredsøe, J., Raaijmakers, T.C., Schouten, J.-J. (2015). Edge scour at scour protections around piles in the marine environment - Laboratory and field investigation. Coastal Engineering, Volume 106, Pages 42-72.
  4. Vousdoukas, M.I., Kirupakaramoorthy, T., Oumeraci, H., de la Torre, M., Wübbold, F., Wagner, B., Schimmels, S. (2014). The role of combined laser scanning and video techniques in monitoring wave-by-wave swash zone processes. Coastal Engineering, Volume 83, January 2014, Pages 150-165.
  5. Tatlock, B., Briganti, R., Musumeci, R.E., Brocchini, M. (2018). An assessment of the roller approach for wave breaking in a hybrid finite-volume finite-difference Boussinesq-type model for the surf-zone. Applied Ocean Research, Volume 73, April 2018, Pages 160–178

Round 2

Reviewer 2 Report

The authors addressed previous comments. The introduction has been greatly improved. The manuscript is acceptable for publication.

Author Response

The authors appreciate your feedback. Thank you.

Reviewer 4 Report

Journal of Marine Science and Engineering

Image-based measurement of wave interactions with rubblemound breakwaters

by Douglas S., Cornett A. and Nistor I.

Reviewer’s report

The paper presents an innovative image processing method developed to track water surface elevation under a wide range of wave conditions.

The authors responded adequately only to part of my previous issues.

Therefore, I suggest reviewing the introduction by paying more attention to the following points:

  • Introduction - Edge detection is only one of the possibilities to given by algorithm of image segmentation. A combination of different image segmentation approaches was used by Musumeci et al. (2013) to track the 3D scour evolution at the toe of a vertical pile.
  • Musumeci,R.E., Farinella,  M., Foti, E.,  Battiato,  S., Petersen, T.U.,   Sumer, B.M. (2013) Measuring sandy bottom dynamics by exployting depth from stereo video sequences.  Lecture Notes in Computer Science (including subseries Lecture Notes in Artificial Intelligence and Lecture Notes in Bioinformatics) Volume 8156 LNCS, Issue PART 1, 2013, Pages 420-430
  • Introduction - As a matter of fact, there are several works which deal with breaking waves. One interesting application of Vousdoukas et al. (2014) to detect wave-by-wave swash zone processes by the combination of LIDAR data and video images.
  • Vousdoukas, M.I., Kirupakaramoorthy, T., Oumeraci, H., de la Torre, M., Wübbold, F., Wagner, B., Schimmels, S. (2014). The role of combined laser scanning and video techniques in monitoring wave-by-wave swash zone processes. Coastal Engineering, Volume 83, January 2014, Pages 150-165.
  • Introduction - The Structure from Motion technique and RGB-D images are now widely used in the field of coastal engineering, by using off-the-shelf tools and algorithms (see Petersen et al. 2015; Musumeci et al., 2018). The authors should critically compare such approaches with their own.
  • Petersen, T.U., Sumer, M., Fredsøe, J., Raaijmakers, T.C., Schouten, J.-J. (2015). Edge scour at scour protections around piles in the marine environment - Laboratory and field investigation. Coastal Engineering, Volume 106, Pages 42-72.
  • Musumeci, R.E., Moltisanti, D., Foti, E., Battiato, S., Farinella, G.M. (2018). 3-D monitoring of rubble mound breakwater damages. Measurement: Journal of the International Measurement Confederation, vol. 117, March 2018, Pages 347-364
